# Complementary Role of Oxytocin and Vasopressin in Cardiovascular Regulation

**DOI:** 10.3390/ijms222111465

**Published:** 2021-10-24

**Authors:** Ewa Szczepanska-Sadowska, Agnieszka Wsol, Agnieszka Cudnoch-Jedrzejewska, Tymoteusz Żera

**Affiliations:** Laboratory of Centre for Preclinical Research, Chair and Department of Experimental and Clinical Physiology, Medical University of Warsaw, 02-091 Warsaw, Poland; awsol@wum.edu.pl (A.W.); acudnoch@wum.edu.pl (A.C.-J.); tzera@wum.edu.pl (T.Ż.)

**Keywords:** oxytocin, vasopressin, hypertension, myocardial infarction, stress

## Abstract

The neurons secreting oxytocin (OXY) and vasopressin (AVP) are located mainly in the supraoptic, paraventricular, and suprachiasmatic nucleus of the brain. Oxytocinergic and vasopressinergic projections reach several regions of the brain and the spinal cord. Both peptides are released from axons, soma, and dendrites and modulate the excitability of other neuroregulatory pathways. The synthesis and action of OXY and AVP in the peripheral organs (eye, heart, gastrointestinal system) is being investigated. The secretion of OXY and AVP is influenced by changes in body fluid osmolality, blood volume, blood pressure, hypoxia, and stress. Vasopressin interacts with three subtypes of receptors: V1aR, V1bR, and V2R whereas oxytocin activates its own OXTR and V1aR receptors. AVP and OXY receptors are present in several regions of the brain (cortex, hypothalamus, pons, medulla, and cerebellum) and in the peripheral organs (heart, lungs, carotid bodies, kidneys, adrenal glands, pancreas, gastrointestinal tract, ovaries, uterus, thymus). Hypertension, myocardial infarction, and coexisting factors, such as pain and stress, have a significant impact on the secretion of oxytocin and vasopressin and on the expression of their receptors. The inappropriate regulation of oxytocin and vasopressin secretion during ischemia, hypoxia/hypercapnia, inflammation, pain, and stress may play a significant role in the pathogenesis of cardiovascular diseases.

## 1. Synthesis and Release of Oxytocin and Vasopressin

Oxytocin (Cys-Tyr-*Ile*-Gln-Asn-Cys-Pro-*Leu*-Gly-NH2; OXY) and arginine vasopressin (Cys-Tyr-*Phe*-Gln-Asn-Cys-Pro-*Arg*-Gly-NH2; AVP) are nonapeptides, differing only in two aminoacids [1,2,3]. In vertebrates, OXY and AVP are synthesized mainly in the hypothalamus as preprohormones, namely preprooxytocin and preprovasopressin. They are then processed along the axonal projections to the posterior lobe of the pituitary, where they are stored in secretory vesicles and released to the peripheral circulation as final products [4]. The neurons secreting OXY and AVP are located mainly in the supraoptic nucleus (SON), paraventricular nucleus (PVN), and the suprachiasmatic nucleus (SCN) [2,5,6,7,8]. In mammals, the preprohormone of vasopressin consists of vasopressin (most frequently arginine vasopressin, in some species lysine vasopressin), neurophysin and glycopeptide copeptin, which are cleaved and released in equimolar amounts. The stoichiometric generation of vasopressin and copeptin enables the measurement of the blood copeptin concentration instead of the AVP concentration [9,10,11,12]. The preproprotein of OXY cleavage results in the final release of oxytocin and its carrier protein, neurophysin [3].

In all species, OXY and AVP genes are located on the same chromosomes. They lie on opposite strands and are transcribed in opposite directions [3]. In human beings, mice, and rats, the genes encoding OXY and AVP prepropeptides are mapped correspondingly to chromosome 20p13 (www.ncbi.nlm.nih.gov/gene/551 (accessed on 15 October 2021)), chromosome 2 (www.ncbi.nlm.nih.gov/gene/11998 (accessed on 21 October 2021)) and chromosome 3 (www.ncbi.nlm.nih.gov/gene/24221 (accessed on 30 September 2021)). Under pathological conditions, the synthesis of inappropriate peptides is caused by polymorphisms of genes encoding specific elements of the OXY-AVP pathways [13]. 

Apart from AVP and OXY, vertebrates can synthesize lysine vasopressin (LVP, some mammals), vasotocin (birds, amphibians, fish), isotocin (teleost fish), and mesotocin (frogs) [1]. These compounds are released either to the systemic circulation and act as potent multifunctional substances or they are released locally and regulate the function of neighboring cells [14,15].

The main source of circulating oxytocin and vasopressin is the neurohypophysis, which is situated in the pituitary fossa of the sphenoid bone together with the adenohypophysis. In mammals, hypophysis is vascularized by the superior hypophyseal artery, which emerges from the circle of Willis. The branches forming the arteriae infundibularis vascularize the infundibular part of the adenohypophysis and form a capillary network on the external surface of the infundibulum, through which blood flows into the sinusoidal capillaries of the anterior lobe [16,17]. Neurohypophysis is pierced by multiple vessels. The inferior hypophyseal artery enters the organ of the posterior circumference of the distal neurohypophysis (neural lobe, posterior lobe). The blood capillaries of the posterior lobe possess a fenestrated endothelium with multiple pores, enabling the local release of peptides into the adenohypophysis and/or the cerebrospinal fluid. In the posterior lobe, the endings of the neurosecretory axons form a dense net together with blood capillaries and glial cells [16,17]. The process of fusion of the secretory vesicle with the membrane and the secretion of peptides involves activation of protein kinase A and protein kinase C and the influx of intracellular calcium. 

Vasopressinergic and oxytocinergic projections also reach several regions of the central nervous system (CNS) and the spinal cord. Both peptides are released from axons, soma, and dendrites and modulate the excitability of other neuroregulatory pathways [15,18,19,20,21,22,23]. Several observations have revealed the release and/or action of OXY and AVP in the olfactory bulb, orbitofrontal cortex, anterior cingulate gyrus, amygdala, striatum, hippocampus, bed nucleus of the stria terminalis, suprachiasmatic nucleus, and the autonomic ganglia [7,15,18,19,24,25,26,27,28,29,30,31,32,33]. The local synthesis and action of vasopressin and oxytocin in the peripheral organs, such as the eye, heart, and gastrointestinal system has been observed (see below and [34,35,36,37,38]).

The secretion of oxytocin and vasopressin is enhanced by hyperosmolality, reduction of the circulating blood volume, hypotension, hypoxia, stress, and hyperthermia [17,39,40,41,42,43,44,45,46]. The effects of these signals are associated with the activation of multiple receptors and transmitting pathways. Neurons synthesizing oxytocin and vasopressin communicate with several neurotransmitter systems and participate in the neuromodulation and regulation of emotions and behavior [18,47,48,49,50,51,52]. The abundant innervation of oxytocinergic and vasopressinergic neurons enables the precise adjustment of the central and systemic secretion of OXY and AVP to specific physiological conditions and changes in the internal and external environment (see below and [34,35,36,37,38]).

## 2. Vasopressin and Oxytocin Receptors

Vasopressin operates through the activation of three subtypes of receptors: V1aR (AVPR1a), V1bR (AVPR1b, V3R), and V2R (AVPR2). Oxytocin acts through its own OXTR receptors, but it can also stimulate V1aR. The AVP/OXY receptors belong to different subtypes of the G protein-coupled receptors (GPCRs). Specifically, V1aR, V1bR, and OXTR belong to the Gq/11 protein family. The Gq/11 protein family participates in the activation of the phospholipase C signaling pathway, in the mobilization of Ca^2+^ from intracellular stores, in the activation of protein kinase type C (PKC), and in the generation of inositol 1,4,5-trisphosphate (Ins3P) and diacylglycerol (DAG). V2R belongs to the Gs protein receptors engaged in the activation of protein kinase A (PKA) and the adenylate cyclase signaling pathway, employing intracellular cAMP as the intracellular second messenger [4,46,53,54,55,56]. The PKA-independent intracellular effects of V2R stimulation on aquaporin 2 (AQP2) in the kidney have also been described [57].

### Location of Vasopressin and Oxytocin Receptors

The genes for OXY and AVP as well as for AVPR1a, AVPR1b, AVPR2, and OXTR are located on different chromosomes [47,58,59,60]. These genes may manifest polymorphisms which are associated with inappropriate responsiveness to the ligand [61,62,63]. In humans, the OXTR gene is present as a single copy, which is mapped to the gene locus 3p25–3 (https://www.ncbi.nlm.nih.gov/gene/5021 (accessed on 17 October 2021)), while AVP receptor genes are mapped to different loci, namely AVPR1a to 12q14.2 (https://www.ncbi.nlm.nih.gov/gene/552 (accessed on 19 September 2021)), AVPR1b to 1q32.1 (https://www.ncbi.nlm.nih.gov/gene/553 (accessed on 19 September 2021)) and AVPR2 to Xq28 (https://www.ncbi.nlm.nih.gov/gene/554 (accessed on 8 July 2021)).

The synthesis and functional engagement of vasopressin V1aR, V1bR, and OXTR in several regions of the CNS as well as in the peripheral organs has been observed. Experiments have made it possible to detect V1aR, V1bR, and OXTR mRNAs and proteins in the cerebral cortex, hypothalamus, pons, medulla, and cerebellum [3,29,33,50,64,65,66,67,68,69,70,71,72]. V1R and OXTR are also present in the arteries, veins, heart, lungs, carotid bodies, kidneys, adrenal glands, pancreas, gastrointestinal tract, eyes, ovaries, uterus, thymus, and autonomic ganglia [30,35,37,38,46,69,73,74,75,76,77,78]. Vasopressin V2R mRNA and protein were found in the kidneys and the urinary bladder [58,79,80]. V2R mRNA was also detected in the human gastrointestinal tract [37]. It may be that V1aR, V2R, and OXTR form dimers or heteromers with other receptors and that this type of processing may have an impact on the responsiveness to the ligand [81,82,83,84,85].

## 3. Regulation of Oxytocin and Vasopressin Secretion

### 3.1. Neuronal and Non-Neuronal Oxytocin Transport and Secretion

The human oxytocin gene (20p13) consists of three exons. The first exon encodes the translocator signal, the oxytocin peptide, the processing signal (glycyl-lysyl-arginine—GRK), and the first nine residues of neurophysin I, the second exon encodes the central part of neurophysin, and the third exon encodes the COOH-terminal region of neurophysin. In most species, the OXY gene extends over 800 bp. Its regulatory region shows remarkable polymorphism [86,87,88]. After synthesis, oxytocin is either transported in large dense-core vesicles to the neurohypophysial terminals in the posterior pituitary or it is stored in the somato-dendritic regions of the magnocellular neurons of the supraoptic and paraventricular nuclei [21,89]. Transport of vesicles depends on the activation of protein kinase A (PKA) and protein kinase C (PKC). The activation of PKA results in an enhanced association of kinesin-2 with annexin A1 (ANXA1) and increased axon-localization of the OXY vesicles. The activation of PKC attenuates the binding of kinesin-2 to ANXA1 and reduces the axonal transport of OXY [90]. Oxytocin is also secreted from the axon terminals into the hypophyseal portal blood circulation of the median eminence [91]. The axonal exocytosis of oxytocin into the portal capillaries involves conventional mechanisms of neuronal exocytosis with the activation of the primary calcium voltage-dependent channels and subsequent intracellular Ca^2+^ influx. This is followed by the formation of a core complex (known as the soluble N-ethylmaleimide sensitive fusion protein attachment protein receptor, SNARE), which enables the fusion of the membrane of the vesicle to the cellular membrane, and the subsequent release of the cargo into the extracellular space [92,93,94,95,96]. The somato-dendritic release of oxytocin also depends on the entry of the extracellular Ca^2+^. The magnocellular neurons express various types of calcium channels (L, P/Q, N, and R) located on the somata and dendrites [97,98,99]. The excitability of the OXY neurons is regulated by several factors operating in the excitatory and inhibitory synapses, such as adhesion molecules, neurotransmitters, regulatory peptides, and gasotransmitters [100,101,102]. It has been shown that the somato-dendritic release of OXY can occur independently of depolarization, and that the dendritically released oxytocin can exert both local autocrine effects to modulate the activity of the mother neurons and paracrine effects on the surrounding neurons and glial cells [100].

It has been established that OXY is synthesized and secreted in the peripheral organs, including the heart, the kidneys, adrenals, testes, and uterus [3,34,103,104]. Radioimmunoassay and molecular studies on rats have revealed a high expression of OXY and OXTR in the cardiomyocytes of the right atrium and in the left and right ventricles [35,105]. OXTRs are also present in human vascular endothelial cells, large vessels, and cardiac microvessels [35,103,106]. The synthesis of oxytocin in cultured beta cell lines originating from murine, rodent, and human pancreatic islets has been observed [36].

### 3.2. Regulation of Oxytocin Secretion

The secretion of OXY is triggered by classical reproductive and maternal stimuli such as birth, maternal care of the offspring, suckling and lactation, and mating and sexual stimulation in males and females. In addition, it is modulated by dehydration, hyperosmolarity, hypovolemia, hemorrhage, and exposure to stressors (Figure 1). Oxytocin neurons are abundantly innervated and the secretion of OXY is stimulated by several neurotransmitters and neuropeptides (α-melanocyte-stimulating hormone—α-MSH; angiotensins, corticotropin releasing factor—CRF, dopamine, glucocorticoids, norepinephrine, serotonin), and inhibited by GABA, orexins, prolactin, and central opioids [52,101,107,108,109,110,111,112].

In estrogen sensitive loci, such as the uterus, hypothalamus, heart, smooth muscles, and mammary glands, estrogen and its receptors are essential for OXY gene transcription and the induction of OXTR. It has been found that ERα-mediated activity modulates OXTR transcription, whereas ERβ regulates OXY mRNA and OXY levels. In addition, treatment with estradiol increases OXY mRNA expression in the brains of wild-type mice, but not in ERβ knockout (ERβKO) male and female mice [113,114,115].

### 3.3. Regulation of Vasopressin Secretion

The synthesis and secretion of vasopressin and oxytocin show a remarkable similarity. In humans, the AVP gene, linked on chromosome 20p13, consists of three exons and encodes the nonapeptide vasopressin, neurophysin II, and copeptin [88,116,117,118]. Under physiological conditions, vasopressin regulates the water balance, body fluid osmolality, arterial blood pressure, body temperature, susceptibility to stress, and emotional challenges, and its secretion is adjusted to fluctuations of these parameters [14,119,120,121,122] (Figure 1). 

Vasopressinergic neurons are particularly sensitive to changes in sodium concentration. The threshold level of sodium concentration for AVP release is estimated at 140 mEqNa^+^/l, and an increase by 1–2% is sufficient to stimulate further release. Other substances increasing osmolality, such as glucose, urea, or ATP are less effective stimulants [8,19,44,116,123,124,125,126]. However, it should be noted that, in the explants of the supraoptic nucleus, the effectiveness of glucose in releasing vasopressin is noticeably potentiated by insulin [124]. It is postulated that the increase of osmolality causes a shrinkage of the hypothalamic magnocellular neurons. This results in cytoskeletal alterations and an opening of the delta-N variant of the transient receptor potential vanniloid (TRPV1) channels, which serve as the molecular detectors of osmotic stress [127,128]. The opening of these channels results in depolarization of the neuronal soma (from −60 mV to −55 mV) which is followed by the generation of action potentials and the release of AVP [14]. TRPV1 activity is potentiated by an influx of calcium ions through L-type Ca^2+^ channels and by an activation of the protein kinase C-dependent pathway [129]. The osmoregulation of AVP secretion can be inhibited by anticipatory stimuli engaging the anterior cingulate area [14]. The hypothalamic vasopressinergic neurons receive a rich input from the neurons located in multiple regions of the brain and the spinal cord and are abundantly furnished with receptors for neurochemical compounds that modulate the sensitivity of these neurons to osmotic stimuli [130,131]. They are also located in the median eminence of the anteroventral third ventricle region (AV3V). The destruction of this region results in the inhibition of AVP secretion, and thus results in hyperosmolality and hypernatremia [132].

The release of vasopressin is also regulated by signals generated in the osmotic receptors of the gastrointestinal tract and liver, which may play an essential role in the anticipative adjustment of AVP secretion to the osmolality of an ingested meal [133,134].

Vasopressin secretion is significantly influenced by changes in blood volume, blood pressure, and oxygen content that are signaled to the vasopressinergic neurons by afferent inputs from the baroreceptors, low pressure mechanoreceptors, and arterial chemoreceptors, respectively [135,136,137]. The neurochemical background of the processes linking the transmission of signals from the cardiovascular receptors and chemoreceptors to the vasopressinergic neurons is still being investigated. The secretion of AVP is under the control of several groups of neurons engaged in the central regulation of blood volume and pressure, such as the rostral ventrolateral medulla (RVLM), the caudal ventrolateral medulla (CVLM) [138], and the circumventricular organs [116].

The mRNA encoding cAMP responsive element-binding protein 3-like 1 (CREB3L1)—a transcription factor of the CREB/activating transcription factor (ATF) family—may play an essential role in the enhancement of AVP transcription in the PVN and SON neurons [139]. Therefore, it is probable that all neuroactive compounds affecting CREB3L1 activity may influence the synthesis of AVP in the hypothalamus. Potent stimulators of AVP release during hypovolemia and hypotension include catecholamines, angiotensin II, and mineralocorticoids [140,141]. It has been observed that atrial natriuretic peptide (ANP) participates in the inhibition of AVP secretion during hypervolemia [50,142,143,144]. 

AVP secretion may be significantly modulated by mineralocorticoids. The SON and PVN neurons express 11β-HSD2 and mineralocorticoid receptor (MR) immunoreactivity, and the neurons immunoreactive for AVP and MR colocalize with those expressing epithelial sodium channels (ENaC). Moreover, it has been shown that the administration of mineralocorticoids induces the upregulation of AVP mRNA [145,146]. Particularly interesting is the interaction of estrogens with vasopressin. Estrogen receptors are expressed in the hypothalamic vasopressinergic neurons and ERβ inhibits the release of vasopressin in the PVN and SON. Because estrogen expression decreases during dehydration and high salt intake, it has been postulated that estrogen availability may have an impact on the regulation of the sensitivity of vasopressinergic neurons to osmotic stress [147].

There is strong evidence that the vasopressinergic system is activated during stress and that the blockade of V1 AVP receptors significantly reduces stress-induced anxiety and depression behavior [147,148,149,150,151,152,153]. Serotonin, Ang II, and cytokines may be engaged in the stimulation of AVP release during various types of stress [44,45,48].

Vasopressin secretion is elevated during hyperthermia, particularly during the heating of the brain thermosensitive region in the preoptic area [154]. In addition, a prominent direct effect of temperature on the temperature-sensitive ion current mediated by the activation of the TRPV1 channels (a six-fold increase in the firing rate of the action potentials) has been demonstrated in vasopressin neurons [155].

## 4. Oxytocin and Vasopressin Secretion during Cardiovascular Disorders

### 4.1. Oxytocin Secretion in Cardiovascular Diseases 

Extensive experimental data have shown that arterial hypertension, myocardial infarction, and heart failure, or their consequences, such as pain and stress, have a significant impact on the oxytocin system. 

Observations on spontaneously hypertensive rats (SHR) and their normotensive counterparts WKY have revealed that SHR has a lower level of oxytocin in the hypothalamus, midbrain, and brainstem, and showed reduced expression of OXY mRNA in the PVN, SON, and the posterior pituitary. They also manifest reduced expression of oxytocin receptors mRNA in the nucleus of the solitary tract (NTS) [156,157,158,159]. Jameson and coworkers reported that the release of OXY from the PVN into the brainstem is reduced in rats exposed to chronic intermittent hypoxia/hypercapnia, and that this effect is associated with the symptoms of human obstructive sleep apnea syndrome (OSAS) and hypertension [27]. 

The secretion of oxytocin is significantly affected in various models of hypertension and heart failure. The activation of the OXY neurons in the PVN prevents the development of arterial hypertension in the animal model of OSAS [27]. Downregulation of the OXTR protein and reduction of OXTR mRNA expression was found in the right ventricle of rats with pulmonary hypertension [160]. Several studies have demonstrated a correlation between cardiac oxytocin system activity and the development of post-infarction heart failure. In rats with confirmed post-infarction heart failure, a higher expression of OXY mRNA and an elevated OXY protein level were found in the right ventricle, whereas the OXTR protein level was high in both the left and right ventricle [105]. Other authors reported the downregulation of OXTR in the cardiac left ventricle during the early post myocardial infarction period. However, with the development of heart failure, the OXTR expression markedly increased [161]. Interestingly, rats with myocardial infarction also expressed a higher level of OXTR protein in the superior cervical ganglion [162], and this finding suggests the involvement of OXY in the modulation of sympathetic transmission after myocardial infarction.

Pain is a frequent symptom of CVD and plays an essential role in the regulation of the oxytocin system [163,164,165,166]. For instance, neuropathic pain significantly upregulates OXY mRNA expression in the SON and PVN neurons [167]. Pain, induced by tissue incision, decreases OXY content in the PVN, but increases its release in the periaqueductal grey (PAG) and the caudate nucleus [168]. Oxytocin reduces the pain intensity [166] and changes in its secretion may play a role in the regulation of sensitivity to pain in CVD.

Stress is also inherently present in CVD. Experimental studies showed that short-lasting stress increases the expression of OXY mRNA in the hypothalamus of rats [169] and decreases expression of OXTR mRNA in the heart of Sprague Dawley and Lewis rats [170]. Normotensive WKY rats receiving short intracerebroventricular (ICV) applications of OXY responded to stress with greater increases in blood pressure [171]; however, the same study showed that the enhancement of the pressor response to stress by exogenous OXT in normotensive WKY rats resulted from the interaction of OXY with AVP V1a receptors [171]. In another study, chronically stressed WKY rats treated with continuous ICV infusions of OXTR antagonist manifested significantly greater increases in blood pressure after the application of acute stress [172].

To date, few researchers have investigated the secretion of oxytocin in cardiovascular disorders in humans. Sivukhina et al. reported no significant differences in OXY-positive neuron expression in PVN and SON in the brains of patients, who died from chronic heart failure [173]. More recent findings on patients undergoing aortic valve replacement surgery revealed that myocardial ischemia is associated with significant changes in the expression of several genes involved in the oxytocin signaling pathway. Importantly, myocardial ischemia suppressed oxytocin signaling more effectively in females than in males [174].

### 4.2. Vasopressin Secretion in Cardiovascular Diseases

In both humans and animals, hypertension and heart failure are associated with an elevated secretion of vasopressin [175,176,177]. This is partly caused by the altered action of signals incoming from the cardiovascular baroreceptors, volume receptors, and chemoreceptors, and partly by other challenges generated by the disease, such as water–electrolyte and acid–base balance disorders, pain, and stress.

In experimental models of arterial hypertension, AVP content and its gene expression are elevated in the PVN, SON, and NTS [178,179]. There is evidence that several components of the renin-angiotensin system (RAS) and mineralocorticoids are involved in the activation of the vasopressin system in hypertension [45,179,180]. The engagement of mineralocorticoids is suggested by studies showing an abundant expression of mineralocorticoid receptors (MR) and α, β, and γ subunits of the ENaC in the cardiovascular brain regions innervated by vasopressinergic neurons, as well as by an upregulation of AVP mRNA and V1aR after the administration of mineralocorticoids [145,180,181]. It is worth noting that mineralocorticoids induce a greater expression of AVP, V1aR, and MR in SHR than in WKY rats [146].

Moreover, experimental data revealed that heart failure is associated with significant changes of AVP content in cardiovascular brain regions. Specifically, rats with heart failure produced by an overloading of the left ventricle manifested significantly elevated AVP levels in the PVN, the SON, the parafascicular nucleus, and the NTS. Simultaneously, they reduced the AVP content in the locus coeruleus. These changes were significantly attenuated by the blockade of AT1 receptors (AT1R) or the inhibition of angiotensin-converting enzyme (ACE), suggesting the engagement of the RAS in the activation of the vasopressinergic system in heart failure [177,182]. 

Among the factors that may enhance the release of AVP in the heart is hypoxia, which stimulates both the central and the systemic components of the vasopressin system [176,183,184]. Hypoxia activates the catecholaminergic neurons of the CVLM projecting to the PVN, and the pressor response evoked from the carotid bodies involves the stimulation of adrenoreceptors in this nucleus [185,186]. Furthermore, the release of AVP in response to hypoxia is mediated by the activation of the carotid body [137]. 

Because pain and stress are strong stimulators of the vasopressinergic system [130,166,169,187,188,189,190,191,192], it is very likely that they play an essential role in the stimulation of vasopressin release in CVD. It has been suggested that the activation of hypothalamic vasopressinergic neurons by pain is regulated by norepinephrine and the stimulation of noradrenergic alpha receptors [192]. 

Patients with cardiovascular pathologies, predominantly heart failure, but also myocardial infarction and arterial hypertension, present elevated plasma AVP levels [175,193]. Population-based data have shown elevated plasma AVP levels and its surrogate marker copeptin [10] in various types of heart failure, acute myocardial infarction, diabetes development, and hypertension [11,12,194,195,196,197]. For this reason, copeptin can be used in clinical practice as a predictive marker in cardiovascular diseases. Corresponding to findings from animal-derived models, higher AVP immunoreactivity in the PVN, the SON, and the posterior pituitary was also found in patients who died from chronic heart failure [173]. 

## 5. Cardiovascular Effects of Vasopressin and Oxytocin

As mentioned above, the secretion of oxytocin and vasopressin is regulated by the same stimuli in many instances and both peptides act through receptors engaging the same intracellular pathways. As emphasized below, they can exert complementary actions, either through intensification or abatement of their own regulatory effects.

### 5.1. Effects of Oxytocin on the Cardiovascular System

Information about the effects of oxytocin on the cardiovascular system has been provided by a number of experimental studies. Oxytocin receptors are present in the heart, in the vessels, and in the cardiovascular regions of the brain [35,109,198,199,200,201]. In the vascular endothelial cells of large vessels and cardiac microvessels, they co-localize with endothelial nitric oxide (NO) synthase (eNOS) [106]. In the estrogen-sensitive regions of the brain, such as the hypothalamus and the amygdala, the induction and transcription of OXTR are regulated by estrogens and depend on the stimulation of estrogen receptors (ER). Stimulation of ERα enhances induction of OXTR, while activation of ERβ helps to maintain the transcription of OXTR [113,202]. 

The majority of experimental findings indicate that the administration of oxytocin exerts a hypotensive effect and that the magnitude of this effect depends on its central and peripheral actions (Figure 2). In normotensive rats remaining at rest, ICV administration of OXY significantly decreases blood pressure, but does not influence the heart rate [203,204]. It has been shown that the subchronic (5 days) subcutaneous application of OXY increases the hypotensive action of clonidine, and this action is abolished by the blockade of alpha-2 receptors (α2R). The latter finding suggests the involvement of α-2 noradrenergic receptors in the hypotensive action of OXY [205]. In Sprague Dawley rats, OXTR mRNA was detected in the vena cava, the pulmonary vein, and the aorta. Its presence was associated with a high expression of ANP mRNA in the vena cava and the pulmonary vein. OXTR mRNA was also identified in the mesenteric arteries and the uterine arcuate arteries of female Sprague Dawley rats. The expression of OXTR mRNA was enhanced by estrogen in the aorta and the vena cava [103,206] and it is worth noting that oxytocin did not cause vasodilation of the systemic arteries. In contrast, in high concentrations, OXY elicited vasoconstriction, which was mediated by the stimulation of V1aR [206].

Transcripts of OXTR were demonstrated both in the atrial and ventricular sections of the heart. It has been shown that activation of the oxytocin receptors plays an essential role in the release of ANP in the heart [34]. In vitro observations on pluripotent P19 embryonic stem cells have shown that OXY induces differentiation of these cells into cardiomyocytes [207].

Extensive findings from animal-derived models indicate that oxytocin may play a beneficial role in the regulation of cardiovascular functions in hypertension. In particular, peripheral subcutaneous injections of OXY in SHR rats resulted in hypotension and caused greater diurnal and nocturnal reductions in blood pressure [208,209]. It appears that the beneficial effect of oxytocin on blood pressure develops during the early postnatal period [210,211]. The buffering role of oxytocin in the regulation of blood pressure in SHR is impaired and even reversed during exposure to an alarming stress [171,172]. SHR rats respond to the application of acute stress with greater increases in blood pressure than WKY rats and this difference is abolished by the chronic ICV administration of OXY [172]. Hypertension is associated with the altered expression of oxytocin in the central nervous system and the adrenal glands of SHR [156]. Reduced expression of OXTR was found in SHR in the NTS and the dorsal brain stem [157,158]. More recently, it was shown that oxytocin released by neurons projecting from the PVN to the dorsal motor nucleus of the vagus prevents the development of hypertension in rats chronically exposed to hypoxia/hypercapnia—an animal model of human hypertension associated with OSAS [27]. 

Some studies suggest the engagement of oxytocin in hypertension-induced cardiac remodeling. In particular, it has been found that the blockade of cardiac OXTR facilitates the development of fibrosis and causes a reduction in the LV ejection fraction in SHR rats [200]. SHR also manifests elevated levels of OXY and OXTR in the left ventricle of the heart [212].

The cardiac oxytocinergic system plays an essential role in cardiac remodeling and cardioprotection during cardiac ischemia and post-infarct heart failure [161,213,214,215,216,217,218,219]. In the isolated heart rat model, submitted to a one hour ischemia-reperfusion-perfusion procedure, the administration of oxytocin during the early reperfusion phase reduced the size of the myocardial infarct and increased the coronary blood flow. This was associated with a decreased production of reactive oxygen species (ROS), a reduced expression of pro-inflammatory cytokines (TNF-α, IL-1β, IL-6), and reduced apoptosis, reduced arrhythmia, and reduced ventricular fibrillation [217,218,219,220]. In addition, OXY increased the phosphorylated Hsp27 protein level and the atrial natriuretic peptide level in the cardiac left ventricle [217]. It should be noted that the cardioprotective actions of OXY are mediated through OXTR activation. The infusion of oxytocin elevated cell death in siRNA-mediated knockout cells with reduced expression of OXTR via binding with vasopressin receptors [220]. Centrally administered OXY decreases both the pressor response and the tachycardic response after exposure to an alarming stress in rats with myocardial infarction, but not in the sham-operated controls [221].

In the rat model of heart failure induced by pressure overloading of the left ventricle lasting four weeks, chronic stimulation of PVN neurons in adult Sprague Dawley rats resulted in several beneficial effects, such as a decrease in blood pressure and heart rate, a reduction of cellular hypertrophy, a reduction of fibrosis, a reduction of IL-1β concentration, and an improvement of the LV function [222]. Gutkowska et al. showed that oxytocin may prevent the development of diabetic cardiomyopathy. A downregulation of OXY gene expression in the heart of the db/db mouse model of type 2 diabetes mellitus has been reported [223]. Male db/db mice respond with a higher expression of cardiac OXTR, a lower fasting blood glucose level, reduced body fat accumulation, reduced ROS production, reduced cardiomyocyte hypertrophy, reduced fibrosis, and reduced apoptosis to prolonged (12 weeks) administration of oxytocin [215,224]. However, chronic overstimulation of OXTR may result in opposite and highly detrimental effects. It has been shown that 14-week-old transgenic α-MHC-Oxtr mice overexpressing oxytocin receptors manifest significantly greater left ventricle end-diastolic volume, prominent cardiac fibrosis, and enhanced expression of pro-fibrogenic genes. There is also a higher mortality rate among transgenic α-MHC-Oxtr mice than among the wild-type control strain [225].

To date, little is known about the cardiovascular effects of OXY in humans and further studies are needed. In line with experimental findings, OXY and its analogue carbetocin exert hypotensive effects during delivery. Rosseland et al. reported marked hypotension and an increase in the stroke volume after OXY/carbetocin administration during cesarean section [226]. Furthermore, in another study, the intravenous infusion of OXY during cesarean section resulted in stenocardia, transient profound tachycardia, hypotension, and electrocardiographic signs of myocardial ischemia [227]. It is worth noting that intranasal OXY administration caused a short-term increase in heart-rate variability in healthy volunteers [228].

### 5.2. Effects of Vasopressin on the Cardiovascular System

Vasopressin regulates blood pressure, blood flow, and body fluid volume through actions mediated by receptors located on vasopressin sensitive cardiovascular neurons in the brain and in the spinal cord, and on AVP receptors in numerous peripheral organs [29,60,69,73,123,125,126,130,175,229,230,231,232,233] (Figure 3). All types of vasopressin receptors are engaged in these actions. The central pressor action of vasopressin is mediated mainly by V1aR located in the RVLM and the rostral ventral respiratory column [234,235,236].

Stimulation of V1aR also plays an essential role in the vasoconstriction of systemic, coronary, and renal vessels [237,238,239,240]. However, stimulation of V1aR also causes vasodilation of the pulmonary arteries [241]. In the kidney, V1aR is present in the basolateral membranes of the thick ascending limbs of Henle’s loop (TAL) and the collecting ducts (CD), and in the juxtaglomerular apparatus [242,243]. Experiments on V1aR deficient mice (V1aR^−/−^) strongly suggest that activation of V1aR in the kidney plays an essential role in the stimulation of the RAS and in the activation of the V2R-AQP2 pathway. Elimination of V1aR in these mice resulted in polyuria and the diminution of renin-containing granular cells and the plasma renin level, as well as in the reduction of V2R expression in the collecting ducts [244,245]. Downregulation of V1a receptors is observed in the vascular smooth muscles in sepsis elicited by the application of lipopolysaccharide or proinflammatory cytokines, and this may account for the reduced responsiveness to vasopressin in septic shock [246].

V1bR is located mainly in the brain, the pituitary, the pancreas, and the kidneys [55,247,248]. In the brain, V1bR mediates the activation of the PVN presympathetic neurons by vasopressin [235,249]. They are also engaged in the stimulation of insulin release from the pancreatic islet cells, and the release of ACTH from the anterior pituitary [250,251]. Expression of V1b receptor mRNA in the anterior pituitary is mediated by glucocorticoids [250]. V1bR is located mainly in the inner medulla in the kidney. There is evidence that stimulation of renal V1bR can counterbalance the antidiuretic effect of vasopressin mediated by V2R [252]. It is likely that the counteractive effects of the stimulation of V1bR and V2 receptors by vasopressin may account for the potent natriuretic effect observed after the combined administration of insulin and vasopressin in rats [253].

Vasopressin has long been known as an antidiuretic hormone due to its water retaining action in the kidney, exerted by V2 receptors. In the renal tubules, V2R is located mainly in the collecting duct cells, where it is coupled to Gs proteins. The stimulation of V2R results in the activation of adenylate cyclase, the formation of cAMP, and the activation of protein kinase A (PKA). This leads to the fast activation of AQP2 molecules and their translocation from the cytoplasmic vesicles to the apical cellular membrane where they form water channels. The channels enable the shift of water from the tubules to the cells and subsequently to the basolateral membrane. PKA also intensifies the synthesis of AQP2 molecules [254,255,256]. Activation of V2R increases water permeability in the kidney and enhances the shift of urea and sodium. Acting on V2 receptors, AVP enhances the transcription of the epithelial sodium channels (ENaC) and urea transporters, and promotes the synthesis of renin [254,255,256,257,258]. Consequently, the stimulation of V2R reduces water and sodium excretion and their blockade results in diuresis and natriuresis. The effectiveness of these actions depends on the state of hydration of the body and the availability of other factors operating in the renal tubules, such as Ang II and aldosterone. Aldosterone cooperates with AVP in the regulation of sodium transport via the epithelial sodium channels in the distal portion of the nephron [259]. Experiments on rats with diabetes insipidus have shown that aldosterone reduces the expression of AQP2 in the inner stripe of the outer medulla and increases free water clearance and creatinine clearance, whereas the mineralocorticoid receptor antagonist, spironolactone, elevates AQP2 expression and decreases creatinine clearance [260]. The stimulation of V2 receptors causes their internalization and recycling. This recycling is associated with increased sensitivity of the collecting ducts to vasopressin. The process of recycling is intensified in cardiac failure induced by the ligation of the left anterior descending coronary artery and may account for increased sensitivity to the antidiuretic action of vasopressin in heart failure [261]. 

Although the kidney is the main target organ expressing V2R, studies exploring the expression of V2R mRNA and the functional consequences of V2R blockade have shown that these receptors may also play a role in the regulation of the gastrointestinal tract, the heart, and the brain [37,58,80,236,262]. Experiments on embryonic stem cells have shown that vasopressin contributes to cardiac differentiation and its action is accomplished by the stimulation of V2 receptors and NO synthesis [263].

The blockade of vasopressin receptors may exert beneficial effects in the treatment of various forms of cardiac failure. The application of antagonists selectively blocking V2 receptors in the rat induces a significant aquaphoric effect and improves hemodynamics [264]. Experiments on pigs with cardiac failure induced by rapid pacing showed that the systemic administration of the V1a receptor antagonist (SR49059) elicited beneficial effects illustrated by a reduction of the left ventricle dimension and peak wall stress, although it did not improve cardiac muscle contractility. The authors reported that the effectiveness of the treatment of heart failure was significantly greater after simultaneous application of V1aR and angiotensin II antagonists [265]. The altered expression of V1aR has been reported in rats with myocardial infarction and in rats exposed to chronic stress. The infarcted rats manifested a reduced V1aR mRNA expression in the preoptic, diencephalic, and mesocenphalopontine regions of the brain. In contrast, the expression of V1aR receptors was significantly elevated in the brain medulla and in the renal cortex of chronically stressed rats [266]. Taking into account the pressor effect of the stimulation of V1a receptors in the brain, it is likely that the downregulation of these receptors during the post-infarct state is a protective response, which is suppressed during stress. There is evidence that the stimulation of V1aR may reduce the cerebral blood flow during dehydration and that this response is associated with an increased production of ROS and can be effectively abolished by the blockade of V1aR [267].

The results from clinical observations are in line with the experimental data. Patients with heart failure manifest a significantly greater expression of V1aR in the left ventricle myocardium than subjects with non-failing hearts [268]. To date, the application of various types of V1aR and V2R antagonists in the treatment of heart failure has provided inconclusive results in human clinical trials [269,270,271]. Nonetheless, recent clinical guidelines on the diagnosis and the treatment of heart failure issued by the European Society of Cardiology indicate that tolvaptan, a selective V2R antagonist, can be considered for the treatment of heart failure patients with hyponatremia [272]. Recently, promising results have been obtained in a double-blind, randomized phase II AVANTI study following the effects of the application of pecavaptan, a dual V1a/V2 antagonist in patients with heart failure and fluid overload. This study revealed that treatment with pecavaptan significantly enhances decongestion of the heart and improves the maintenance of the body fluid balance in patients with acute heart failure [273].

## 6. Conclusions

The secretion of oxytocin is elevated by reproductive and maternal stimuli and by hyperosmolality, hypovolemia, hemorrhage, pain, and stress. The process of stimulation is related to the action of physiochemical factors, neurotransmitters (catecholamines, serotonin), neuropeptides, glucocorticoids, and estrogens. The secretion of oxytocin is inhibited by GABA, orexins, prolactin, and opioid peptides.The secretion of vasopressin is stimulated by hypernatremia and the activation of TRPV1 channels, and by hypovolemia, hypotension, hypoxia, hyperthermia, pain, and stress. The stimulation is mediated by neurotransmitters (catecholamines, serotonin), neuropeptides (ANG II, cytokines), and mineralocorticoids. Hyponatremia, hypervolemia, and estrogens inhibit the secretion of vasopressin.The cardiovascular system is regulated jointly by oxytocin and vasopressin and their action can be synergic or antagonistic.Oxytocin exerts a hypotensive action, increases the production of vasodilatory compounds (ANP, NO), decreases pressor responses to hypoxia and stress, and exerts several cardioprotective effects in hypertension, cardiac ischemia, and cardiomyopathies. During post-infarct heart failure, oxytocin reduces the production of ROS and inflammatory cytokines, decreases apoptosis, decreases arrhythmia, decreases ventricular fibrillation, and reduces cardiovascular responses to stress.Vasopressin increases the retention of water and sodium and exerts a pressor effect, which is mediated by the stimulation of the sympathetic nervous system and direct vasoconstrictive action exerted in the majority of the vascular beds. It also interacts with pressor and hypotensive compounds (angiotensins, mineralocorticoids, NO, ANP). Cardiovascular responses to vasopressin at rest and during stress are enhanced in hypertension and heart failure. Experimental and clinical studies show that the blockade of vasopressin receptors may exert beneficial effects in patients with heart failure and fluid overload.To summarize, the present review emphasizes the important role of oxytocin and vasopressin in the regulation of the cardiovascular system. Vasopressin and oxytocin play complementary roles in the regulation of cardiovascular parameters, exerting synergic and antagonistic effects in the brain, in the vessels, and in the kidney. Evidence from experimental observations exploring the role of oxytocin and from experimental and clinical studies analyzing the role of vasopressin strongly suggests that the application of either oxytocin or vasopressin antagonists may exert favorable effects in cardiovascular diseases and extensive clinical studies are necessary to confirm this conclusion.

## Figures and Tables

**Figure 1 ijms-22-11465-f001:**
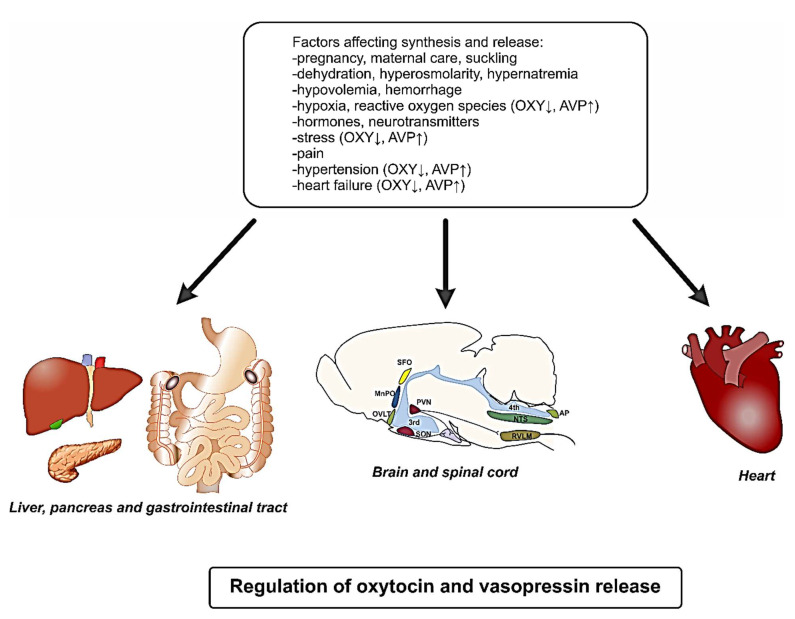
Main factors operating in the brain and peripheral organs participating in the regulation of the synthesis and release of oxytocin and vasopressin. Abbreviations: AP—area postrema; AVP—arginine vasopressin; MnPO—median preoptic nucleus; NTS—nucleus tractus solitarii; OVLT—organum vasculosum of the lamina terminalis; OXY—oxytocin; PVN—paraventricular nucleus; RVLM—rostral ventrolateral medulla; 3rd—third ventricle; 4th—fourth ventricle.

**Figure 2 ijms-22-11465-f002:**
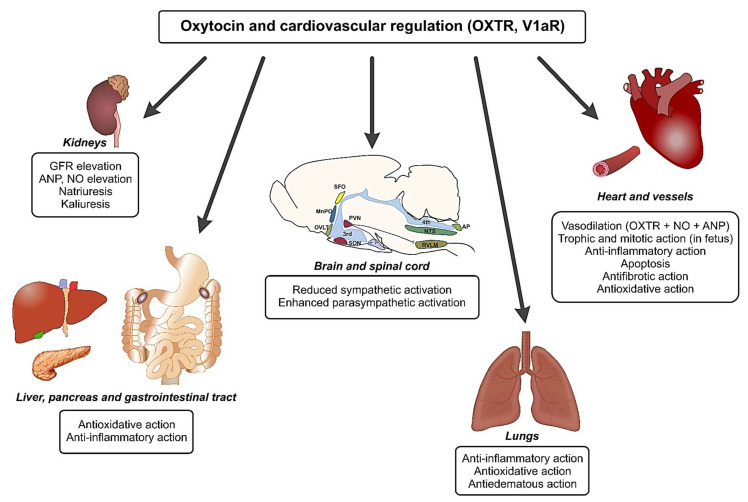
The role of oxytocin in cardiovascular regulation. Oxytocin stimulates specific receptors (OXTR) and vasopressin receptors (V1aR) in different organs. In the central nervous system, it regulates the activity of several groups of neurons and may exert either stimulatory or inhibitory effects, depending on the site of action. The predominant effects of the action of oxytocin in the central nervous system involve reduced activation of the sympathetic division of the autonomic nervous system and enhanced activation of the parasympathetic division of the autonomic nervous system. Abbreviations: AP—area postrema; MnPO—median preoptic nucleus; NTS—nucleus tractus solitarii; OVLT—organum vasculosum of the lamina terminalis; PVN—paraventricular nucleus; RVLM—rostral ventrolateral medulla; 3rd—third ventricle; 4th—fourth ventricle.

**Figure 3 ijms-22-11465-f003:**
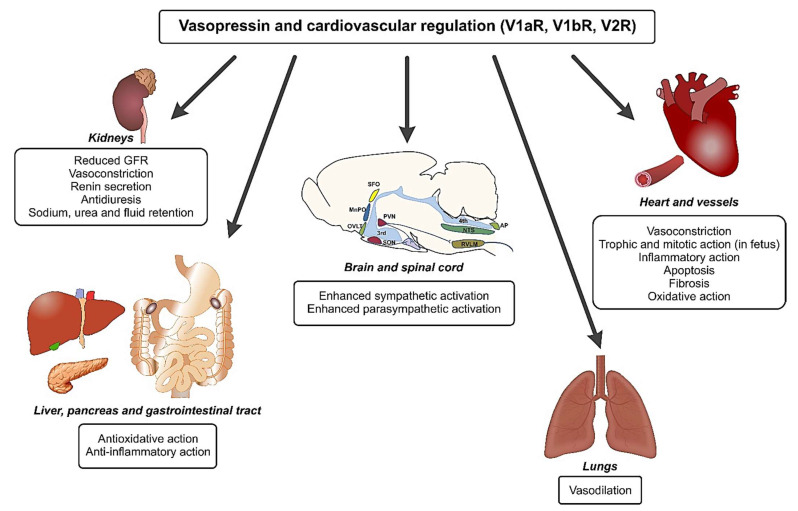
The role of vasopressin in cardiovascular regulation. Vasopressin stimulates specific types of receptors (V1aR, V1bR, and V2R) in various organs. In the central nervous system, it regulates the activity of several groups of neurons and exerts either stimulatory or inhibitory effects, depending on the site of action. Consequently, the effects of activation of both divisions of the autonomic nervous system (sympathetic and parasympathetic) can be observed, depending on specific factors provoking activation of the vasopressinergic system. Abbreviations: AP—area postrema; MnPO—median preoptic nucleus; NTS—nucleus tractus solitarii; OVLT—organum vasculosum of the lamina terminalis; PVN—paraventricular nucleus; RVLM—rostral ventrolateral medulla; 3rd—third ventricle; 4th—fourth ventricle.

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
