# Peer review of "Complementary Role of Oxytocin and Vasopressin in Cardiovascular Regulation"

_ijms, 2021, doi:10.3390/ijms222111465_

Round 1

Reviewer 1 Report

The article entitled “Complementary role of oxytocin and vasopressin in cardiovascular regulation” by Ewa Szczepanska-Sadowska et al. submitted for publication in ijms, is a review article which reported various effects of oxytocin and vasopressin. The article is well documented but written in a mediocre style, repetitions of words and sentences are frequent (evidence: 27, studies: 19) and is poorly illustrated. The three figures are similar and not informative. A few grammatical errors should be corrected.

The authors mixed results obtained in rat, frog, bird, cell culture, human. It will be logical to separate the different information according to the experimental model or the human studies.

The summary at the end of the article did not fit with the text and a conclusion will be more appropriate.

The number of pages of references is above that of text. Are all the references useful (for instance line 112, line 114)?

Author Response

POINT-BY-POINT RESPONSES TO GENERAL AND SPECIFIC COMMENTS

Reviewer No 1:

We would like to thank  for the critical opinion, helpful comments and suggestions. We hope that the changes in the txt of the manuscript are satisfying and you find manuscript acceptable for publication. For your convenience, we have highlighted the changes in the text and included the edited segments in our response.

Reviewer No 1:The article is well documented but written in a mediocre style, repetitions of words and sentences are frequent (evidence: 27, studies: 19) and is poorly illustrated. The three figures are similar and not informative. A few grammatical errors should be corrected. The authors mixed results obtained in rat, frog, bird, cell culture, human. It will be logical to separate the different information according to the experimental model or the human studies

It was our intention to provide maximum information in possibly concise manner. Details concerning species would expand text. In the revised version we introduced more information about species on page 3 (lines 105, 113, 119), page 4  (lines 154), page 6 (line 186), page 16 (line 545). We also corrected some sentences to avoid repetitions and the grammatical errors that appeared in the text. Figures were also formatted according to the Reviewer’s No. 2 suggestions.

Reviewer No 1:The summary at the end of the article did not fit with the text and a conclusion will be more appropriate.

According to the Reviewer’s suggestion we introduced “Conclusions” to our article and excluded Summary section.

Reviewer No 1:The number of pages of references is above that of text. Are all the references useful (for instance line 112, line 114)?

The literature concerning oxytocin and vasopressin is abundant and we selected the most important references. We would prefer to keep all of them.

Reviewer 2 Report

 The review by Szczepanska-Sadowska and colleagues explores the role of oxytocin and vasopressin in cardiovascular regulation.

The review is well structured and reports on the evidence, mostly derived from animal studies on the cardiovascular involvment of oxytocin and  vasopressin.

I have some comments:

-Line 160: I suggest changing the term “stimulated” with “modulated” to include the negative stimulation on oxytocin secretion exerted by some of the stimuli (eg. Stress).

-Line 249: eliminate “the” before arterial hypertension and myocardial infarction, this applies to the following occurrences of pathology names.

-Line 262: prevents

-line 546: please reformulate to a better English form“or being its consequence, such as pain and stress.”

- cardiovascular disease should be abbreviated as CVD throughout the text

-Figure A1 would be more useful and clear if the superior section was dedicated to the different stimuli affecting OXY and AVP secretion with the lower part illustrating the target secreting organs.

The sections on OXY and AVP secretion in cardiovascular disease include evidence from both human and animal studies, often alternating between the two. It would be more appropriate to separate the available evidence into only human and only animal studies

-Lines 279-284 the association between pain and cardiovascular disease is only superficially addressed, there is no specification of the type of nociception involved and if the cited studies were conducted in animal models or in humans.

-In the section regarding AVP and copeptin secretion in CVD the authors could address the emerging role of copeptin as a marker of cardiovascular risk, eg. in early rule out of acute coronary syndrome. Some references: https://ccforum.biomedcentral.com/articles/10.1186/s13054-020-02904-8, https://www.hindawi.com/journals/dm/2015/614145/

Minor observations:

-Figures are indicated as Figure A1, A2 and A3, while in the text they are Figure 1,2,3. Is there a reason for this?

-The names of the authors are reported both in the extended form and as abbreviations. Please correct.

-Figures 2 and 3 could have better formatting: ie “Lung” is not centered with the picre above, in addition, the space between tiles and text boxes is not always the same.

Author Response

POINT-BY-POINT RESPONSES TO GENERAL AND SPECIFIC COMMENTS

Thank You for all of the critical suggestions. We hope that the changes in the txt of the manuscript are satisfying and you find manuscript acceptable for publication. For your convenience, we have highlighted the changes in the text and included the edited segments in our response.

Reviewer No 2:Line 160: I suggest changing the term “stimulated” with “modulated” to include the negative stimulation on oxytocin secretion exerted by some of the stimuli (eg. Stress).

Thank you for suggestion. Correction - line 163

Reviewer No 2:Line 249: eliminate “the” before arterial hypertension and myocardial infarction, this applies to the following occurrences of pathology names.

Thank you for suggestion. Correction - line 258

Reviewer No 2:Line 262: prevents

Thank you for suggestion. Correction – line 271

Reviewer No 2: line 546: please reformulate to a better English form“or being its consequence, such as pain and stress.”

Thank you for suggestion. Correction line 259

Reviewer No 2:cardiovascular disease should be abbreviated as CVD throughout the text

In the text “cardiovascular diseases” are abbreviated as CVD. We left full term in titles of subsections of our manuscript.  

Reviewer No 2:Figure A1 would be more useful and clear if the superior section was dedicated to the different stimuli affecting OXY and AVP secretion with the lower part illustrating the target secreting organs.

Fig 1 was edited according to the Reviewer’s suggestion.

Reviewer No 2:The sections on OXY and AVP secretion in cardiovascular disease include evidence from both human and animal studies, often alternating between the two. It would be more appropriate to separate the available evidence into only human and only animal studies

We introduced information concerning species on the following pages: page 3 (lines 105, 113, 119), page 4  (lines 154), page 6 (line 186), page 16 (line 545). We wanted to avoid very detailed information because this would expand the text.

Reviewer No 2:Lines 279-284 the association between pain and cardiovascular disease is only superficially addressed, there is no specification of the type of nociception involved and if the cited studies were conducted in animal models or in humans.

The role of oxytocin in the regulation of pain has been recently discussed in several recent  articles and we refer to them on page 8 lines 289, 290, 292.

Reviewer No 2:In the section regarding AVP and copeptin secretion in CVD the authors could address the emerging role of copeptin as a marker of cardiovascular risk, eg. in early rule out of acute coronary syndrome. Some references: https://ccforum.biomedcentral.com/articles/10.1186/s13054-020-02904-8, https://www.hindawi.com/journals/dm/2015/614145/

We emphasize the usefulness of estimations of copeptin on page 2 (lines 39-40), page 8 (lines 320-323) We have also introduced reference 12:

Reinstadler, S.J.; Klug, G.; Feistritzer, H.J.; Metzler, B.; Mair, J. Copeptin testing in acute myocardial infarction: ready for routine use? Dis Markers. 2015; 2015:614145. doi: 10.1155/2015/614145. 

Reviewer No 2:Minor observations:

Figures are indicated as Figure A1, A2 and A3, while in the text they are Figure 1,2,3. Is there a reason for this? The names of the authors are reported both in the extended form and as abbreviations. Please correct. Figures 2 and 3 could have better formatting: ie “Lung” is not centered with the picture above, in addition, the space between tiles and text boxes is not always the same.

The corrections of all indicated fragments were made. They appeared during final edition of text.

Figures 1-3 were also graphically formatted. Missing arrow to “Lungs” at Fig 2 was added. 

Manuscript was also corrected by native speaker.

Round 2

Reviewer 1 Report

The reviewer is sorry to see that beside few modifications in the text the authors, as they stated in the response, did not take into account the points raised by reviewer 1.

The two reviewers clearly indicated that it is not rational to mix evidences obtained in human, in animal and in experimental models. The reviewer two strongly suggested that the authors had to separate the information in different sections. The authors add some sentences but did not follow the advices of the reviewers.

In the cover letter authors wrote that a “native speaker” revised the article, there is no significantly improvement.

The new version could not be considered to be suitable for publication in the present form.

Author Response

POINT-BY-POINT RESPONSES TO GENERAL AND SPECIFIC COMMENTS

Reviewer No 1:

It was not our intention to give the impression of not following the reviewers’ valuable suggestions and comments. We introduced a “conclusion” section at the end of manuscript and added detailed information about different species in the previous version of our manuscript.

In the present version, in sections 4 and 5 where we discuss changes in AVP and OXY secretion and their effects in cardiovascular diseases, we have separated the experimental data from human studies (see page 8, lines 295-299, page 9 – lines 317-324, 327, page 10 lines 362-370, page 14 lines 481-489, page 17 lines 587 and page 18 - Conclusion No 6). The information from human studies was moved to the end of each section, supplemented by additional data and presented in separate paragraphs. 

In Fig.2 Va1R was changed to V1aR.

The manuscript was corrected and edited by a professional English editor.

All changes were marked up using the “Track Changes” function (in both Word and pdf files).

We hope that, with these changes, you will find the manuscript acceptable for publication. Thank you again for your suggestions and the critical review.

Round 3

Reviewer 1 Report

The authors improved the manuscript. They made minimal changes and the article can probably be better, it is their choice. No major pitfall.